# FNBP1 Facilitates Cervical Cancer Cell Survival by the Constitutive Activation of FAK/PI3K/AKT/mTOR Signaling

**DOI:** 10.3390/cells12151964

**Published:** 2023-07-29

**Authors:** Jun Zhang, Xin Li, Yunfei Zhou, Mingming Lin, Qianying Zhang, Yunhong Wang

**Affiliations:** Basic Medical School, Chongqing Medical University, Chongqing 400016, China

**Keywords:** FNBP1, cervical cancer, cell adhesion, FAK/PI3K/AKT signaling, cell survival

## Abstract

Cervical cancer is the most prevalent *gynecological tumor* among women worldwide. Although the incidence and mortality of cervical cancer have been declining thanks to the wide-scale implementation of cytological screening, it remains a major challenge in clinical treatment. High viability is one of the leading causes of the chemotherapeutic resistance in cervical cancers. Formin-binding protein 1 (FNBP1) could stimulate F-actin polymerization beneath the curved plasma membrane in the cell migration and endocytosis, which had previously been well defined. Here, FNBP1 was also demonstrated to play a crucial role in cervical cancer cell survival, and the knockdown of which could result in the attenuation of FAK/PI3K/AKT signaling followed by significant apoptotic accumulation and proliferative inhibition. In addition, the epidermal growth factor (hrEGF) abrogated all the biological effects mediated by the silencing of FNBP1 except for the cell adhesion decrease. These findings indicated that FNBP1 plays a key role in maintaining the activity of focal adhesion kinase (FAK) by promoting cell adhesion. The activated FAK positively regulated downstream PI3K/AKT/mTOR signaling, which is responsible for cell survival. Promisingly, FNBP1 might be a potential target against cervical cancer in combination therapy.

## 1. Introduction

Cervical cancer is a major gynecologic health problem, which is considered the third most widespread tumor in women [1,2,3,4,5]. The prevalence of this fatal ailment is emerging gradually across the globe. In 2020, more than 0.6 million women were diagnosed with cervical cancer [6]. In recent decades, an array of fundamental advances has been made in the diagnosis and treatment of cervical cancer. Both the morbidity and mortality have dropped remarkably thanks to the wide-scale implementation of cytological screening [7,8,9]. However, platinum-based chemotherapy has been the backbone treatment in advanced cervical cancer for quite a few years, with no attractive improvements in survival [10]. Although the combination of carboplatin, paclitaxel and bevacizumab have currently become the standard frontline treatment for cervical cancer with encouraging efficacy [11,12], the search for new strategies that can fulfill the urgent need for effective targeted therapeutic approaches remains a major challenge.

Formin-binding protein 1 (FNBP1), an important member of the F-Bar protein family, consists of a C-terminal Src homology 3 domain (SH3) and an N-terminal region of the extended FER-CIP4 homology domain (EFC), capable of strongly binding and deforming the plasma membrane. The SH3 domain can recruit WASP, WASP-interacting protein (WIP) and dynamin-2 to the plasma membrane for triggering actin polymerization. Both domains are separated by a consensus Rho-binding motif (RBD) that can interplay with Rho proteins [13,14,15]. FNBP1 had been well documented to be extensively involved in cell motility and endocytosis [16,17,18]. However, emerging evidence indicates a novel role of FNBP1 in malignant tumor invasion and metastasis [19,20,21,22,23,24,25]. Here, FNBP1 was confirmed to play a crucial role in maintaining constitutive FAK/PI3K/AKT/mTOR survival signaling in cervical cancer cells. Given that the AKT/mTOR signaling pathway is ubiquitously responsible for cell survival in tumor cells [26,27,28], the new discovery implied a potential target against cervical cancer.

## 2. Materials and Methods

### 2.1. Antibodies and Reagents

The primary antibodies, including antihuman FNBP1 (ab100918), FAK (ab40749, EP695Y), FAK (phospho Tyr 397, ab81298, EP2160Y), PI3K/p85 (phospho Tyr 607, ab182651; ab191606, EPR18702), AKT (phospho Tyr 308, ab38449; ab185633, EPR17671), FOXO1 (phospho Ser 253, ab131339; ab179450), mTOR (phospho Tyr 391, ab32028; ab2732) and GAPDH(ab9485), were all obtained from Abcam (Cambridge, UK). The generalized horseradish peroxidase-conjugated secondary antibodies were obtained from Santa Cruz Biotechnology (sc-2750) (Santa Cruz, CA, USA). The human recombinant epidermal growth factor (rhEGF) was obtained from MCE (Shanghai, China). Sc79, a selective activator for AKT, was provided by Abcam. The specific inhibitors, including Y15 (for FAK) and pictilisib (for PI3K/p110), were obtained from SelleckChem (Houston, TX, USA). The endonucleases used in molecular cloning were obtained from Takara (Dalian, China), while the T4 ligase and reverse transcriptase M-MLV were obtained from Promega (Madison, WI, USA). The Annexin V-FITC early apoptosis detection kit was obtained from Cell Signaling Technology ( Danvers, MA, USA), and the Cytoselect^TM^ cell adhesion assay kit was obtained from Cell BIOLAB, Inc (San Diego, CA, USA). The in vitro toxicology assay kit and propidium iodide (PI) solution were obtained from Sigma-Aldrich (St. Louis, MO, USA). The cross-link Co-IP kit was obtained from Thermo Fisher Scientific (Waltham, MA, USA). The Annexin V-FITC early apoptosis detection kit was obtained from CST (#6592) ( Danvers, MA, USA). The basement membrane extract (BME) was obtained from TianGEN (Matrigel^®^) (Shanghai, China). The other chemical reagents were all AR grade from China.

### 2.2. Cell Culture and Viability Assay

HeLa cells were cultured in RPMI 1640 (Hyclone) supplemented with 15% FBS (Hyclone) and penicillin/streptomycin in a 37 °C humidified incubator containing 95% air and 5% CO_2_. In viability assay, the cells were seeded in 96-well microplates at a density of 1 × 10^4^/mL. The primary cervical cancer cells were separated from the clinical specimens conforming to the standard protocol. Small pieces of tumor tissue (2–4 mm^3^) that were surgically collected from untreated cervical cancer patients, following the written informed consents, were obtained from the subjects. All of the clinical specimen collections were in accordance with the Declaration of Helsinki and the ethics guidelines of the World Health Organization (WHO) and approved by the Ethical Committee of Chongqing Medical University (ECCMU No. 20210107). The tumor tissue pieces were digested with 0.2% type I collagenase (Gibco, Grand Island, NY, USA) in a gas bath thermostatic oscillator at 37 °C and 200 rpm for 40 min. The isolated cells were seeded into 96-well microplates that were precoated overnight with 2 µg/cm^2^ of collagen type I from mouse tail (Sigma-Aldrich, St. Louis, MO, USA). Cells were cultured in keratinocyte serum-free medium (K-SFM) (Gibco). Identification of these primary culture cervical cancer cells were conducted as previously described [29,30]. Cell viability was determined with the in vitro toxicology assay kit (Sigma), according to the manufacturer’s protocol. Each experiment was repeated at least three times.

### 2.3. Construct of shRNA Plasmids and Transfection

Three pairs of oligodeoxynucleotides individually mapping on the different FNBP1 code regions were designated and then chemically synthesized, with each bearing the endonuclease Bam H Ⅰ, Sal Ⅰ and Hind Ⅲ sites at the termini. The corresponding sense/antisense sequences are displayed in the Appendix A, to which those of the chemical synthesized dsRNA were identical. Both the pGenesil-1 plasmids and oligodeoxynucleotides were digested with BamHⅠ/HindⅢ and then ligated together by the T4 ligase (Promega). The same dual cleavages and Sanger sequencing for the shRNA plasmids were used to identify the constructs (Appendix A). The HeLa cells were transiently transfected with the shRNA vectors using the Lipofectamine 3000 reagent (Life Technologies, Carlsbad, CA, USA), abiding by the user’s manual, and then harvested 24 h after transfection. The scrambled shRNA was taken as the negative control in the experiments.

### 2.4. Quantitative RT-PCR (qPCR)

The total RNA was isolated from the cultured HeLa cells using the Trizol reagent (Takara, Dalian, China), and 2 μg of RNA were then converted into cDNA with reverse transcriptase M-MLV (Promega), according to the protocol [31]. After reverse transcription, SYBR Green Supermix with ROX (BioRad, Hercules, CA, USA) was used to perform quantitative RT-PCR. The PCR amplification conditions were set as follows: predenaturation at 94 °C for 3 min followed by denaturation at 94 °C for 30 s, annealing at 60 °C for 30 s, elongation at 72 °C for 30 s with repeated 34 cycles and 72 °C for an additional 10 min to repair the termini of the fragments upon the thermal cycle cessation. GAPDH was amplified using the forward primer 5′-GGGAAACTGTGGCGTGAT-3′ and reverse primer 5′-GAGTGGGTGTCGCTGTTGA-3′. FNBP1 was amplified using the forward primer 5′-CTCTGGGATCAGTTTGACAACTT-3′ and reverse primer 5′-TGCCCTGCGTAATCATTCATT-3′.

### 2.5. Western Blot

The cells were seeded in 100 mm dishes at a density of 1 × 10^5^/mL. The protein samples were prepared conforming to the standard procedure and then separated by 12% SDS-PAGE with 30 μg/lane samples loaded. The target proteins were probed with the specific primary antibodies (mAbs) following the electrophoretic transfer onto polyvinylidene fluoride (PVDF) membranes. The blots were immunodetected by enhanced chemiluminescence (ECL, Pierce) after incubation with horseradish peroxidase-conjugated secondary antibodies. The ECL images were acquired using GelDoc XR (Bio-Rad). Quantification of the Western blots was performed using Image J (NCBI).

### 2.6. Hematoxylin–Eosin (HE) Staining

The cells were cultured on the coverslips (1.5 cm × 1.5 cm) at a density of 1 × 10^4^/mL. The samples were air-dried at room temperature for 15 min followed by a standard procedure for HE staining [32,33]. The sample images were acquired under an inverted microscope (objective magnification 40/0.65, Leica). The cell contact area on the substrate was automatically calculated by the image acquisition system.

### 2.7. Cell Proliferation

The cancer cells were plated in 96-well microplates at a density of 1 × 10^4^/mL. The numbers of live cells were evaluated with the cell viability that could be quantitatively read by the XTT assay. The experiments at each time point (the 0, 1st, 2nd, 3rd, 4th and 5th day) were replicated at least three times.

### 2.8. Cell Cycle Analysis with Flow Cytometry

The cells were seeded in 100 mm dishes at a density of 1 × 10^5^/mL. The cultured cells in dishes were washed with sterile PBS solution three times and then digested with 0.25% trypsin for 5 min at 37 °C. The cell suspensions were stained with propidium iodide (Sigma-Aldrich) for cell cycle analyses on a FACS Calibur (B&D, Franklin Lakes, NJ, USA) after fixation by 70% ethanol in the final concentration. The culture time ranged from 1 to 3 days.

### 2.9. Cell Apoptosis Analysis

The cells were plated in 6-well plates and then rapidly pretreated in a logarithmic growth period according to the standard protocol. Apoptosis was measured by flow cytometry (FCM) after dual staining with FITC-conjugated Annexin V and propidium iodide. The procedure was briefly described as follows. The cells were collected by centrifugation and washed with ice-cold PBS. After washing, the cells were resuspended in 1 × Annexin V Binding Buffer at a concentration of 5 × 10^5^ cells/mL. Then, an aliquoted 96 µL of the cell suspension was added into an assay tube with 1 µL of Annexin V-FITC conjugate and 12.5 µL of Propidium Iodide (PI) solution added. The cells were incubated on ice in the dark for 10 min. After incubation, the cell suspensions were diluted to a final volume of 250 µL/assay with ice-cold, 1 × Annexin V Binding Buffer. Then, the samples were immediately analyzed by FCM (B&D). Each experiment was repeated at least three times.

### 2.10. Cell Adhesion Assay

The cell adhesion assay was immediately performed according to the manufacturer’s instructions after the cancer cells were plated in 48-well microtiter plates at a density of 1 × 10^5^/mL.

### 2.11. Co-Immunoprecipitation (Co-IP)

The culture cells were seeded in dishes and then lysed in the lysis buffer (1% Triton X-100, 50 mM HEPES, 150 mM NaCl, 1 mM MgCl_2_, 1 mM EGTA, 10% glycerol, and protease inhibitor cocktail (Roche, Madison, WI, USA)) for 1 h at 4 °C. Cell lysates were harvested and centrifuged at 14,000 rpm for 10 min at 4 °C. An aliquot of the supernatant was collected for analysis. The total protein was quantified using the Bicinchoninic Acid assay (Pierce) and then normalized to a minimum concentration of 1 mg/mL. For Co-IP, 5 μL of the primary antibody was added to the cell lysates and then incubated on a shaking platform for 3 h at 4 °C, which was followed by the addition of 40 μL of Protein A/G Sepharose (Pierce) just before a further 1 h incubation at 4 °C. The immunoprecipitates were sedimented and then washed extensively with the lysis buffer as well as PBS. Proteins were eluted from the beads by heating to 95 °C with 5% SDS sample buffer before Western blot detection.

### 2.12. Bioinformatic and Statistical Analyses

The molecular interaction network was visualized using Osprey 1.2.0, and GO enrichment analysis was then performed. The online tools from UALCAN (https://ualcan.path.uab.edu/) (accessed on 28 May 2023) were utilized for pan-cancer analyses of cervical cancer. All the experiments were performed at least three times, with the data presented as mean ± SD. The statistical software Origin 6.0 for Windows was employed in the data process and analyses. The Student’s *t* test and one-way ANOVA were used for comparing the group means, and the *p*-values ≤ 0.05 were accepted as significant (*), *p*-values ≤ 0.01 as extremely significant (**).

## 3. Results

### 3.1. The Biological Effects of FNBP1 Knockdown on Cervical Cancer Cells

The target shRNA can potently downregulate the expression of FNBP1 at both mRNA and protein levels in culture cells (Figure 1A). The silencing of FNBP1 was shown to significantly reduce the cell confluency and contact areas with increased floating and spherical cells and concomitantly suppress the cell proliferation through causing S-phase arrest and elevating the apoptotic ratio (Figure 1B–E). FNBP1 knockdown also dramatically attenuated cell adhesion (Figure 1F). Further results demonstrated that the FAK/PI3K/AKT/mTOR signaling axis was repressed by the silencing of FNBP1 (Figure 2A). Although the affinity of phosphorylated PI3K/p85 with FAK was lowered, FNBP1 could not directly associate with FAK (Figure 2B). It was therefore inferred that FBNP1 might keep the continuous stimulation to FAK in a circular manner. According to previous insights [34,35,36], FNBP1 has the ability to recruit the N-WASP/WIP complex to the membrane by its conserved acidic residues near the SH3 domain. The spatial positioning of N-WASP/WIP in proximity to the membrane enables the trigger of actin polymerization that facilitates cell spreading. This increases the engagement of integrin with the extracellular matrix, which might lead to potential activation of FAK. These findings suggest that FNBP1 plays a critical role in regulating cellular processes and may have implications in various biological contexts.

### 3.2. FNBP1 Konckdown Mediated Biological Effects Relieved by AKT Reactivation

The activator sc79 enhanced the phosphorylation level of AKT in culture cells, capable of binding to the pleckstrin homology domain (Figure 3A). The reactivated AKT abrogated the cell proliferation inhibition and then negated the apoptosis accumulation and S-phase arrest (Figure 3B,C). However, AKT reactivation could not rescue the impaired cell adhesion activity (Figure 3D). It was assumed that AKT/mTOR signaling axis could not regulate cell adhesion, but cell survival.

### 3.3. Reactivation of FAK/PI3K/AKT/mTOR Signaling by Addition of Exogenous EGF

The phosphoinositide-3 kinase (PI3K)/AKT survival signaling is an important pathway stimulated by epidermal growth factor (EGF) that binds to its receptor (EGFR, a receptor tyrosine kinase). EGF restored PI3K/AKT/mTOR signaling and then abrogated the silencing effects except for drops in cell adhesion and contact areas (Figure 4A–F). Interestingly, EGF/EGFR signaling could bypass FAK autoinhibition through leading to direct phosphorylation of FAK tyrosine 397 in HeLa cells (Figure 4A). Even so, the cell adhesion activity and contact areas could not be resumed by the reactivation (Figure 4B,F). Only recovery of FNBP1 expression from silencing could rescue the cell adhesion activity and contact areas in HeLa cells (Appendix A). FAK inhibition by Y15 could not abolish EGF-invited PI3K/AKT/mTOR signaling reactivation that could be negated by pictilisib, a specific inhibitor of PI3K/p110 (catalytic subunit of PI3K) (Figure 4A). Although FAKI could attenuate the rescue in cell apoptosis by EGF (Figure 4C), FAKI could not abolish EGF-mediated restoration in cell proliferation, cycle and adhesion (Figure 4D–F). Nevertheless, PI3KI could negate EGF-induced recovery in cell proliferation and S-phase arrest (Figure 4D,E). The outcomes indicated that reactivation of AKT was not dependent on FAK, but PI3K in the EGF/EGFR signaling pathway due to immediate activation of PI3K at the same time. Furthermore, activated FAK and PI3K by EGF did not restore cell adhesion activity and contact areas in the milieu. The results suggested that FNBP1 expression is essential to cell adhesion and is closely linked with FAK signaling cross talk.

To validate the findings, the primary culture cervical cancer cells isolated from three clinical specimens with different pathological classification were further investigated. The same results were experimentally acquired (Figure 5), which indicated that FNBP1 played a ubiquitous role in cervical cancer cell survival by FAK/PI3K/AKT signaling. Promisingly, FNBP1 would be among these important target genes in cervical cancer treatments. The chemosensitization of cervical cancers using traditional drugs in combination with the silencing of FNBP1 were assessed, the outcomes of which guaranteed FNBP1 a promising target for cervical cancers in combination therapy (Figure 6).

## 4. Discussion and Conclusions

The PI3K/AKT signaling is fundamental for tumor cell survival. Many survival factors can suppress cell apoptosis by activating AKT anchored to the membrane, which subsequently phosphorylates and inactivates components of the apoptotic machinery. In the pathway, the most important upstream regulator of AKT is PI3K that is usually overexpressed in cervical cancer cells [37]. PI3K can phosphorylate phosphatidylinositol 4,5-bisphosphate (PIP2) and convert PIP2 into phosphatidylinositol 3,4,5-trisphosphate (PIP3), which is required for AKT activation. Inactive AKT can be recruited to the cell membrane by its PH domain firmly associating with PIP3 together with phosphoinositide-dependent kinases 1 (PDK1), which immediately phosphorylates AKT Tyr308 to initiate AKT activation.

FAK acts as a major signaling hub of the focal adhesion (FA) and is actively involved in cell adhesion and survival dealing with the dynamic assembly/disassembly of FAs. Upon integrin engagement with the extracellular matrix (EMC) propelled by the actin polymerization in cell adhesion, paxillin is phosphorylated at Tyr31, Tyr118, Ser188 and Ser190 and then recruited FAK to the FA complex and promoted the autophosphorylation of Tyr397, which thus activates FAK to start a PI3K/Akt survival signaling cascade. Our data demonstrated that FAK could modulate PI3K/AKT signaling in the presence of FNBP1 in HeLa cells. It has previously been well established that FNBP1 participates in cell motility and endocytosis by sharing a common molecular step to coordinate membrane curvature, Rho activities with actin polymerization. Although never before described, the mechanism could be adopted to account for the role of FNBP1 in HeLa cell adhesion and survival. FNBP1 probably facilitated the engagement of integrin with EMC by propelling F-actin assembly in cell adhesion and inevitably participated in PI3K/AKT survival signaling by resultantly activating FAK (Figure 7). FNBP1 is thus convinced to prompt cell adhesion and then trigger the constitutive activation of the FAK/PI3K/AKT/mTOR signaling axis. The new discovery greatly expands the functional repertoire of FNBP1 and suggests a potential target for combination therapy against cervical cancers [38,39,40].

Although the expression of FNBP1 was significantly reduced due to an increased promoter methylation level in the cervical cancers compared with the normal samples, high FNBP1 expression correlated with a better prognosis and survival because of enhanced cell adhesion and prevented the malignant cells from metastasis (Figure 8). Genes with higher expression in malignant tumors were commonly referred to as oncogenes, while those with lower expression were labeled as tumor suppressors [41,42,43]. It is not always the case. FNBP1 was identified as more of a housekeeping gene than a suppressor in cervical cancers, because further knockdown of FNBP1 by RNAi greatly inhibited the cancer cells’ survival.

Many emphases were placed on oncogenes or genes with high expression in cancers, but little attention has been given to nononcogenes or genes with low expression in the past few decades [44,45]. Conversely, FNBP1 knockdown led to significant cell proliferation inhibition by repressing survival signaling. FNBP1 could be a universal anticancer target regardless of expression levels in different types of cancer. The investigation demonstrated that downregulated genes in tumor tissues could be candidate targets and despite the fact that much attention has previously been paid to the upregulated genes. It is time to alter the conventional prejudice. In addition, the study also implicated that it is not only the genetic context but also the expression level that is involved in defining the role of a specific gene. The seeming contradiction that silencing of FNBP1 in vitro appeared to be antitumor, whereas cervical cancer patients who had comparatively higher levels of FNBP1 presented better survivals, could be rationally interpreted relying on the functional transition advanced by the changed expression levels. FNBP1 expression levels were comparatively elevated, which might also promote cancer cell survival in the primary site but simultaneously inhibit cell dissemination. The clinical outcome was determined by interplay between the two opposite sides and ultimately demonstrated better prognosis and survival in cervical cancers.

## Figures and Tables

**Figure 1 cells-12-01964-f001:**
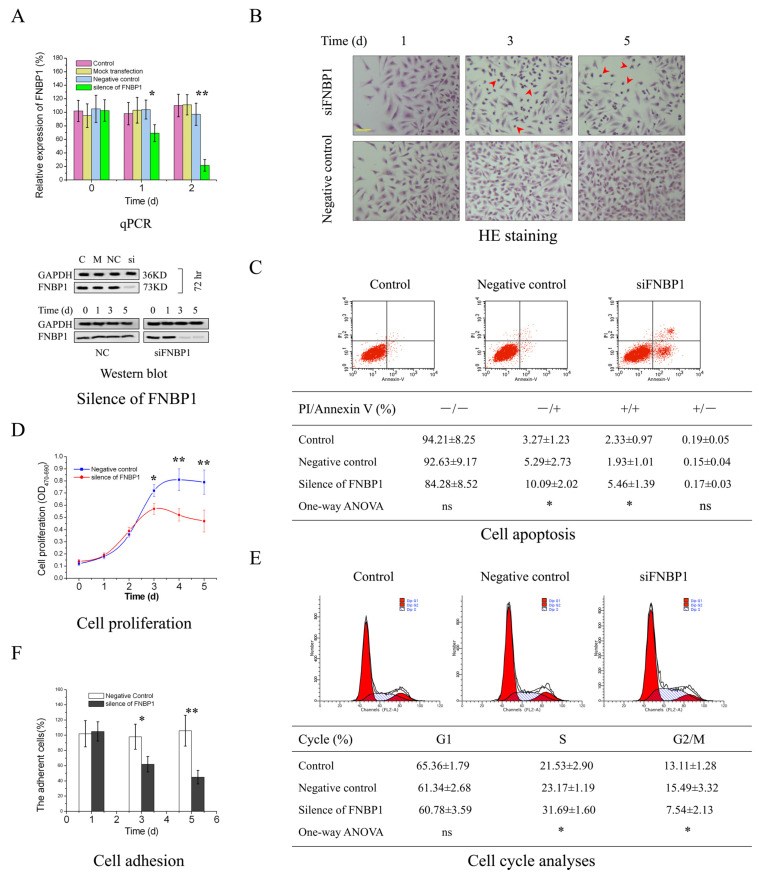
The silencing of FNBP1 in HeLa cells. (**A**) The expression of FNBP1 was screened at both mRNA and protein levels following silencing. Quantification of the blots were in Appendix A. (**B**) The analyses of cell morphology. The yellow scale bar in the figure represented 37.5 μm (400×), and the red arrows indicated the affected cells. (**C**) The apoptotic rates were determined with PI-Annexin V-FITC staining by the flow cytometry (FCM) 3d after silencing. (**D**) The cell proliferation was measured using the XTT assay. (**E**) Cell cycle was analyzed with FCM 3d after silencing. The number of cells enrolled in the analyses was at least 0.5 million. (**F**) Cell adhesion activities were quantitatively evaluated by cell viability. The silencing of FNBP1 was abbreviated as si/siFNBP1. NC represented the negative control. * *p* < 0.05; ** *p* < 0.01; ns, not significant.

**Figure 2 cells-12-01964-f002:**
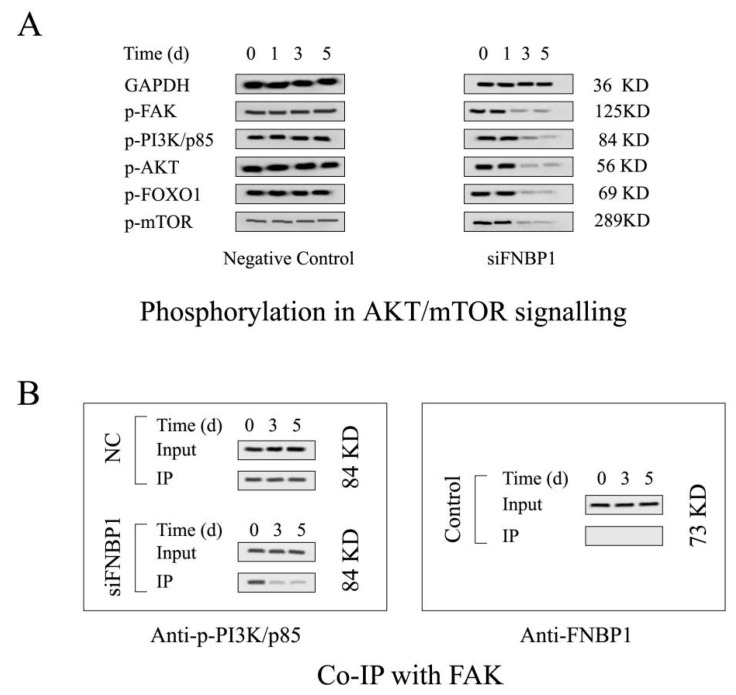
The FAK/PI3K/AKT survival signaling activity was downregulated by the silencing of FNBP1. (**A**) The phosphorylation of the nude proteins in FAK/PI3K/AKT survival signaling axis. (**B**) Co-immunoprecipitation (Co-IP). The total FAK acted as the bait protein. The co-immunoprecipitation of phosphorylated PI3K/p85 and FNBP1 had been detected using Western blotting compared with the input. Quantification of the blots are displayed in Appendix A.

**Figure 3 cells-12-01964-f003:**
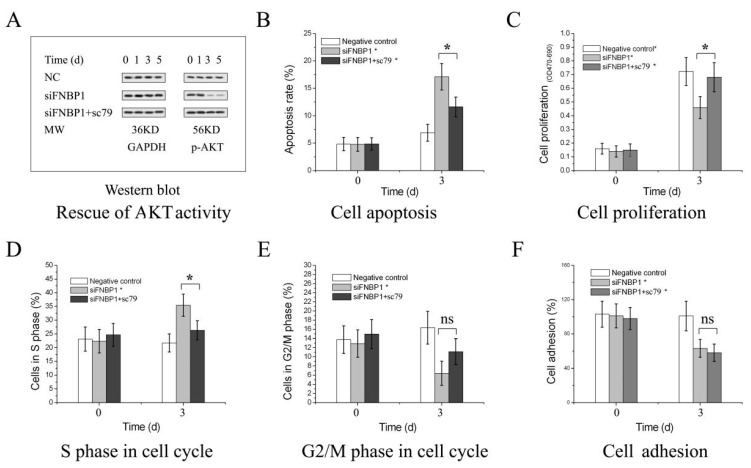
The inhibition of AKT activity was relieved by the specific activator. (**A**) The phosphorylation of AKT Tyr 308. Quantification of the blots were in Appendix A. Analyses of cell apoptosis, proliferation, cycle and adhesion followed by the reactivation of AKT with sc79 were displayed in (**B**–**F**), respectively. The abbreviation of ns referred to no significant difference among the groups. The counted apoptotic cells included both the early [PI(−) Annexin V(+)] and late ones [PI(+) Annexin V(+)]. The compound of sc79 is a potent specific AKT activator. * *p* < 0.05; ns, not significant.

**Figure 4 cells-12-01964-f004:**
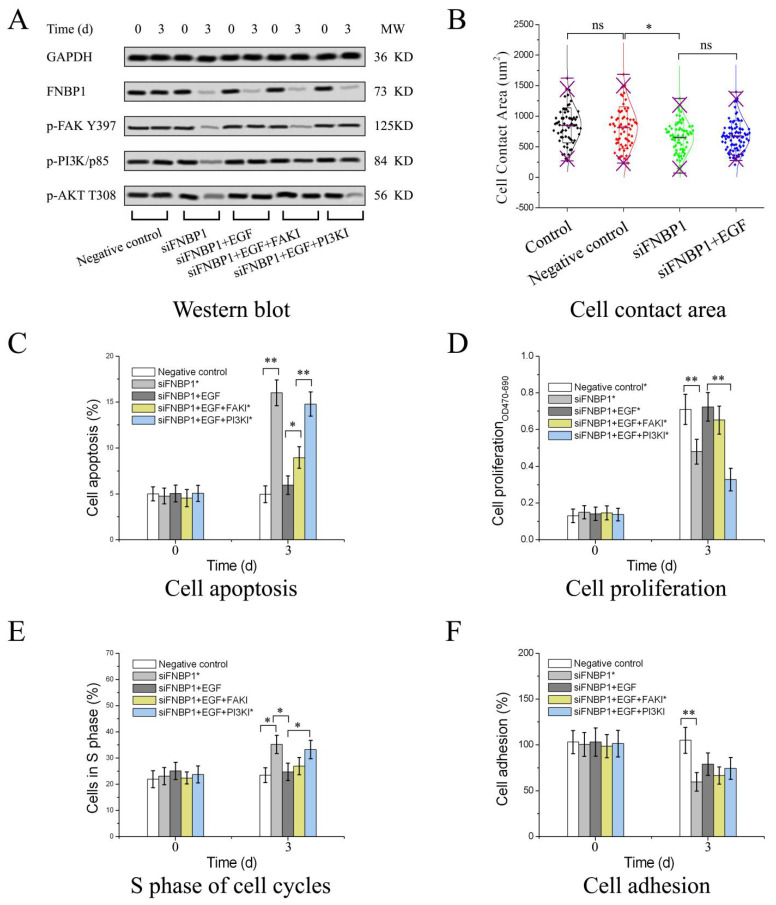
The suppression of FAK/PI3K/AKT signaling was abrogated by EGF. (**A**) The phosphorylation was rescued by the addition of EGF, which could be concomitantly negated by the selective inhibitors. Quantification of the blots are displayed in Appendix A. (**B**) The cell contact area could not be rescued by the addition of EGF. The cell apoptosis, proliferation, S-phase arrest and cell adhesion restored by EGF addition has been presented in (**C**–**F**) individually. Before the addition of EGF, the cells were moderately harvested. The specific inhibitor of FAK (FAKI) is Y15 for blocking FAK autophosphorylation at Y397. The inhibitor of PI3K/p110 (PI3KI) is pictilisib, a potent selective inhibitor targeted to the PI3K catalytic subunit. * *p* < 0.05; ** *p* < 0.01; ns, not significant.

**Figure 5 cells-12-01964-f005:**
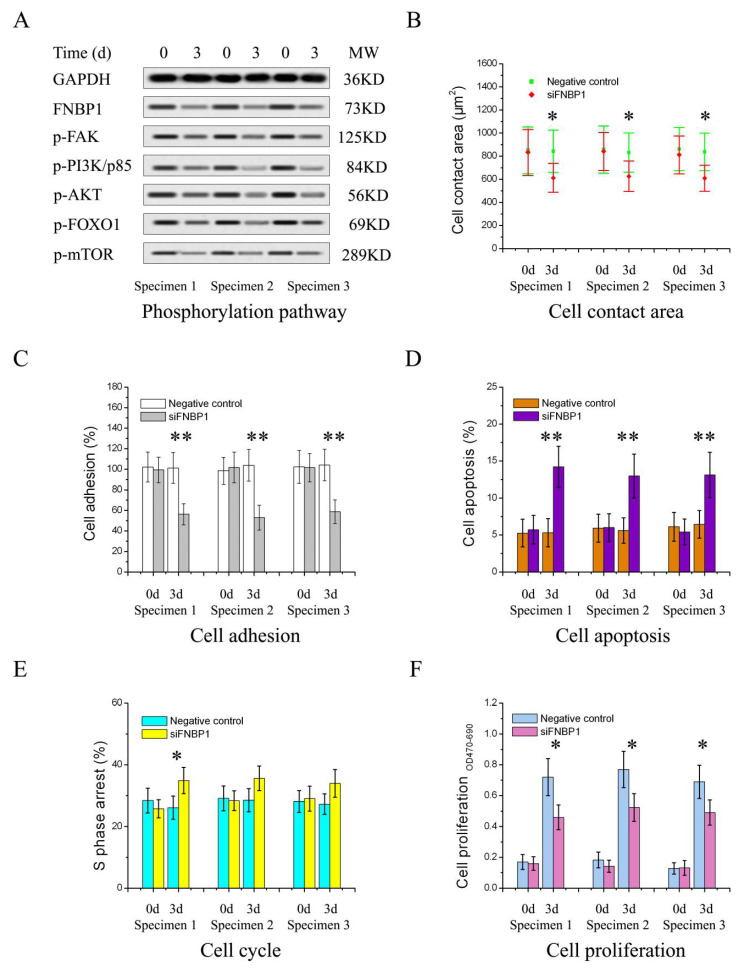
FNBP1 was knocked down in the primary cultured cervical cancer cells separated from the clinical specimens. (**A**) The phosphorylation of the node proteins. Quantification of the blots are displayed in Appendix A. (**B**) The cell contact area. (**C**) Cell adhesion capability. (**D**) Cell apoptotic rates. (**E**) Cell cycle analysis for S-phase arrest. (**F**) Cell proliferation. Specimen 1: cervical adenocarcinoma, IA; Specimen 2: cervical adenosquamous carcinoma, IA; Specimen 3, cervical squamous carcinoma, IB. All the specimens were acquired after informed consent. * *p* < 0.05; ** *p* < 0.01.

**Figure 6 cells-12-01964-f006:**
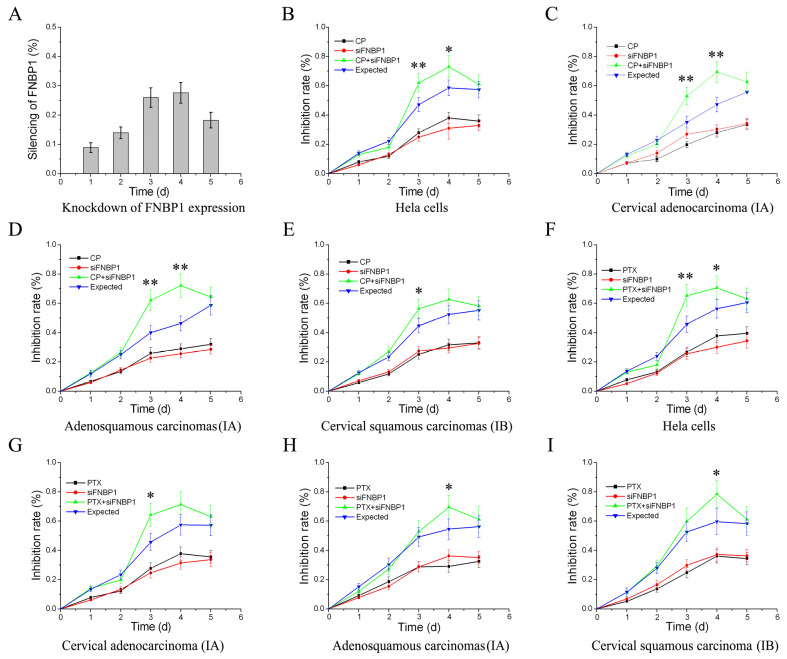
The chemosensitization of cervical cancer by carboplatin or paclitaxel in combination with small interfering RNA targeting FNBP1 (siFNBP1). (**A**) Downregulation of FNBP1 mRNA by RNAi in HeLa cells. Quantification for the silencing of FNBP1 was calculated as follows: the silencing of FNBP1 = 1 − FNBP1si/FNBP1 control. The chemosensitization of cervical cancer by siFNBP1 in combination with carboplatin (CP) are shown in (**B**–**E**), respectively. Those with paclitaxel (PTX) are shown in (**F**–**I**), respectively. The proliferation inhibition by the drug or siFNBP1 was determined as follows: Idrug or siFNBP1 = 1 − ODtreatment/ODcontrol. The expected addictive toxicity of the drug in combination with siFNBP1 was calculated as follows: expt. = 1 − (1 − Idrug) × (1 − IsiFNBP1). The combination can only be considered as chemosensitization if the toxicity observed was significantly greater than the expected. * *p* < 0.05; ** *p* < 0.01.

**Figure 7 cells-12-01964-f007:**
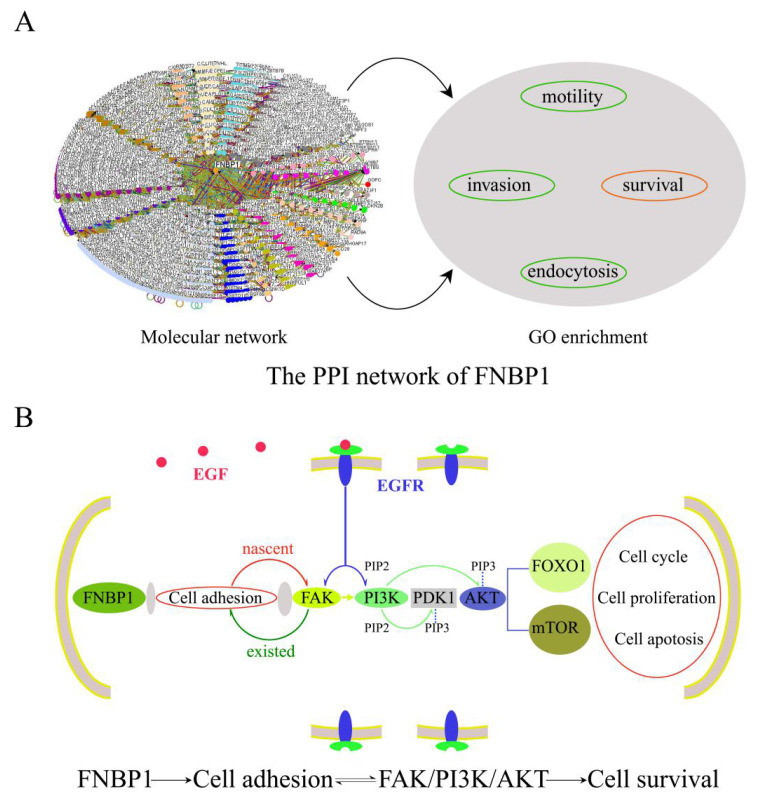
A schematic illustration of the role of FNBP1 in cell survival. (**A**) The molecular network included 1169 proteins and 2074 interactions, and GO enrichment analysis had revealed function modules relating to cell motility, invasion, endocytosis, and survival. The function modules in green had been experimentally validated, while FNBP1’s role in cell survival was confirmed in this investigation. (**B**) The illustration demonstrated that FNBP1 was thought to keep FAK/PI3K/AKT/mTOR survival signaling constitutively activated by promoting cell adhesion efficiently.

**Figure 8 cells-12-01964-f008:**
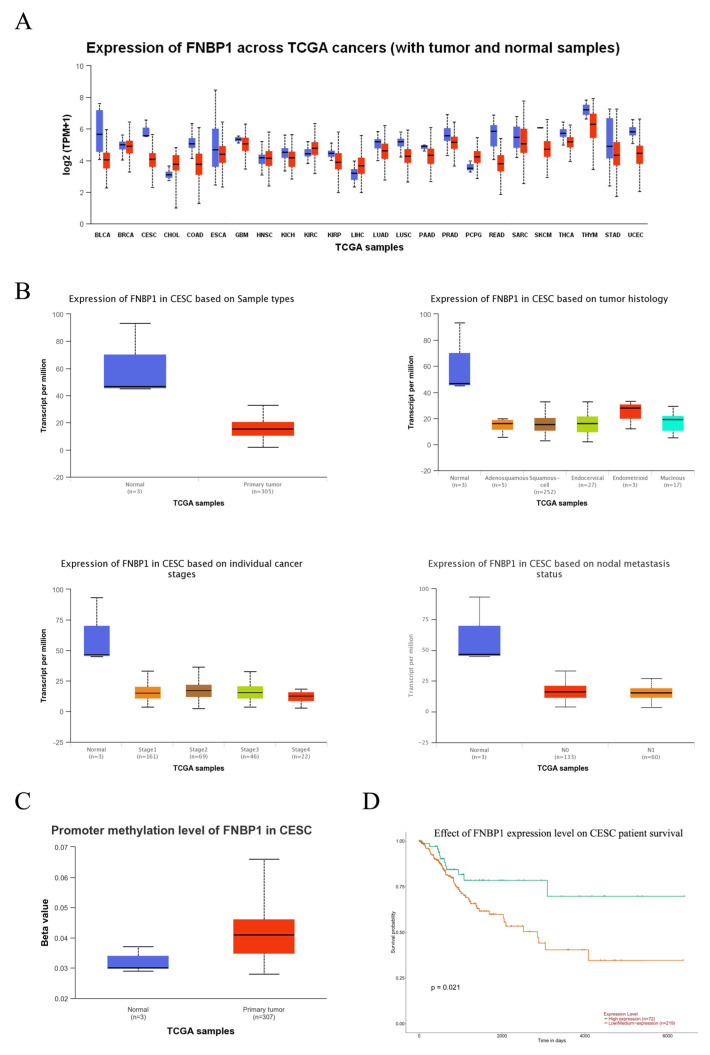
Analysis of FNBP1 in cervical squamous cell carcinoma (CESC). (**A**) The expression of FNBP1 across TCGA cancers. The expression levels of FNBP1 were indicated to vary significantly in different types of cancer (red) compared to the normal tissues (blue). (**B**) Expression of FNBP1 in CESC. In cervical cancers, FNBP1 expression was remarkably downregulated because of promoter methylation. (**C**) Promoter methylation level of FNBP1 in CESC. The promoter methylation level in the cervical cancers increased significantly in contrast with the normal tissues. (**D**) The effect of FNBP1 expression level on CESC patient survival. Relatively higher expression in patients contributed to much slower progression.

## Data Availability

The datasets generated during and/or analyzed during the current study are available from the corresponding author on reasonable request.

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
