# Peer review of "FNBP1 Facilitates Cervical Cancer Cell Survival by the Constitutive Activation of FAK/PI3K/AKT/mTOR Signaling"

_cells, 2023, doi:10.3390/cells12151964_

Round 1
Reviewer 1 Report
I read an interesting paper entitled FNBP1 facilitates cervical cancer cell survival by the constitutive activation of FAK/PI3K/AKT/mTOR signaling. Despite interesting outcomes, the authors did not shy away from some shortcomings, which I will list in points:
v7 – cervical cancer is not 3rd in women. The top three – breast, colorectal and lung cancers – contributed 44.5% of all cancers (excluding non-melanoma skin cancer). Cervical cancer was the fourth most common cancer in women, contributing 6.9% of the total number of new cases diagnosed in 2020.
V31-33 - The sentence is not true. The basis of cervical cancer treatment is surgery then radiation therapy. Radiochemotherapy is also important, but chemotherapy is certainly not the backbone of treatment.
V74 and next - „of 1×104” – error
V75 - Describe the procedure for deriving the cell lines, describe the tumors from which they were derived, and describe how the cancer component was isolated/characterized from the mixture of originally isolated cells. Information as to the consent of the bioethics committee and tissue donor patients (not for publication but for cell line isolation) should be completed in the text (M&M section). Such scans should also be available to the editors.
V117-120 - The procedure is not fully understood. It seems that it may be described incorrectly. If the slides were stained/fixed, why was the microscope inverted and what was "the substrate"?
Reviewer 2 Report
The study focused on Formin binding protein 1 (FNBP1), which plays a crucial role in various cellular processes such as cell migration and endocytosis. The findings revealed that FNBP1 is involved in cervical cancer cell survival, and its suppression leads to a reduction in FAK/PI3K/AKT signaling, resulting in increased apoptosis and inhibition of proliferation. Notably, the effects of FNBP1 silencing, except for decreased cell adhesion, were reversed by epidermal growth factor (hrEGF). These results emphasize the importance of FNBP1 in maintaining focal adhesion kinase (FAK) activity and regulating downstream PI3K/AKT/mTOR signaling, crucial for cell survival. Targeting FNBP1 holds promise as a potential approach for cervical cancer treatment, particularly in combination therapy.
One minor point is to detect the total protein level of each phosphorylated protein in the western blot to prove that the knockdown of FNBP1 didn't affect the total protein abundance.
Reviewer 3 Report
Cells 2482852 -peer review report
General considerations
First, the reviewer wishes to express their gratitude to the editors for giving them the opportunity to help the authors and to the authors for sharing their research.
I also wish to commend the authors for having included STR data for the HeLa cells and full pictures of all three replicates of all the Western blots included in the manuscript.
The manuscript deals with the role of a formin, FNBP1, in the proliferation and survival of cervical cancer cells. The study is conducted both in lab cell line HeLa and in primary patient-derived cells and supplemented with bioinformatic analysis of protein-protein interactions and gene expression in the TCGA cancer database.
The study shows that knockdown of FNBP1 reduces cervical cancer cell viability, increases apoptosis and decreases cell adhesion. This is accompanied by reduction in the phosphorylation of FAK, and in of the PI3K/Akt/FOXO signalling axis. The authors also show that EGF signalling restores phosphorylation and survival but not adhesion and that FAK is not involved in Akt phosphorylation downstream of EGFR. Finally, the authors show that FNBP1 knockdown and chemotherapy show more-than-addictive effects upon combination in both HeLa cells and patient-derived primary cells.
This seems to suggest that intervening on FNBP1 expression and/or function could provide an attractive strategy for therapy of cervical cancer, however the proposed strategy seems at odds with the TCGA data which suggests that increased FNBP1 expression is associated with longer survival.
Major Issues
1- Discrepancy between patient survival and experimental data and its impact on the significance and usefulness of the findings
The TCGA finding that FNBP1 expression in cervical cancer patients is reduced through promoter methylation and that increased FNBP1 expression is associated with significantly longer survival is at odds with the proposed mechanism of pro-oncogenic survival and proliferation explored in the rest of the paper, whereby the authors show that FNBP1 supports tumour cell survival and proliferation through the PI3K/Akt/FOXO signalling axis. The genomic data make it look like a tumour suppressor, while the experimental data shows a behaviour more alike to an oncogene.
This discrepancy is slightly worrying in the context of new therapeutic opportunities, especially in the light of the finding that FNBP1 also supports cell adhesion and that this might be the mechanism through which increased expression of FNBP1 could increase survival by reducing or preventing metastatization through effects on the cytoskeleton and focal adhesions – as mentioned in the comment to Fig 8.
I think this discussion needs to be further expanded with references – there are none in the section at lines 301-304 – and the risk of pursuing a combination therapy against a protein which is a putative anti-metastatic needs to be addressed.
2- Bioinformatics methods and data sources are absent
There is no explanation of the methods and data sources through which Fig7A and Fig 8 have been produced. Which data sources were used for Fig 7A? How was the GO analysis performed? With which software or online service? What was the cutoff chosen for significance and inclusion?
The same questions apply to the data in Fig.8. Which version of the TCGA database was used? How was the data plotted? Which statistics and cutoffs were employed?
Additionally, the data visualisation in Fig 7A is not comprehensible and the caption does not add any information on how to interpret it. I would strongly suggest to re-do the left panel of Fig7A with fewer data and a less busy layout so it can be actually read.
3- Data not shown in results section 3.3
Some data is mentioned regarding re-expression of FNBP1 in silenced Hela cells and its effect on recovery of cell adhesion. Please add another panel to Fig 4, if necessary and show the data.
Additionally, the various panels of Fig 4 are poorly referenced, which makes the argument harder to follow than it should be. Please reference figure panels following statements regarding the data.
Finally, legibility of the concluding statement for section is low. I suggest: “The results suggest that FNBP1 expression is essential to cell adhesion and is closely linked with FAK signalling cross-talk”, or something along these lines.
Minor Issues
1- Details missing from Antibodies and Reagents Section
In the interest of reproducibility, I would strongly encourage the authors to provide the clone number and product code of all antibodies used in the manuscript, as well as the dilution at which they were used.
2- Standard protocols
In methods sections 2.2, 2.6, 2.9 the authors reference standard protocols without further details. In the interest of reproducibility, and because “standard protocols” can vary significantly between institutions, I would strongly suggest the authors to either write it out briefly or to refer to previous literature for each of them.
3- Number of cells per experiment
Please add the starting cell density or seeding density and culturing time to methods sections 2.5, 2.6, 2.8, 2.9 and 2.10.
4- Microscopy details
Please add the details of the objective magnification and NA of the microscope used to take the images in section 2.6. Also, provide details of the coverslips used for the experiment.
5- WB quantification
As all the blots have been done in triplicate with GAPDH normalisation control, I would strongly suggest that the authors perform densitometry on their blots (perhaps using ImageJ) and add panels with the quantification of protein expression changes normalised to GAPDH expression. This would facilitate the assessment of protein expression changes which may be hard to judge by eye.
6- Cell alterations in Figure 1B
In order to facilitate understanding, I would suggest that the alterations in cell confluence and shape discussed at lines 182-184 are pointed out in the figure using arrows/arrowheads.
7- Discussion of IP in figure 2B
I don’t quite understand what point the authors are trying to make about the interaction of FNPB1 and FAK. That it is an indirect interaction? Can the authors please explain better?
8- Fig 4B and Fig 5B data visualisation
Can the authors please clarify what is the data range represented by the box+ whisker plot in Fig 4B? Also, can they clarify whether each point in the plot represents an observation from an individual cell?
Additionally, can the authors explain why, if the experiment was conducted in an analogous way, the visualisation of the data in Figure 5B is different? Can they be harmonized?
9- Fig 6A data – which sample does this refer to?
the manuscript doesn’t explain which cell sample (HeLa or patient-derived cells) was used for the quantification of FBNP1 silencing. Please add this explanation
While overall the readability of the manuscript is good enough to understand the science, there are some issues throughout:
a. phosphorylation is consistently misspelled
b. knockdown is used both as a noun and as a verb (the verb should be knocked down)
c. silence is used in place of silencing throughout
d. the determinate article “the” is used where it shouldn’t in many places.
e. there are some verb tense issues between past and present.
f. There are some misspellings in the axes of Fig S1B, Fig 1A, Fig 6A
Round 2
Reviewer 3 Report
Dear Authors,
thank you so much for making amendments to your work based on my comments.
I am happy with all the edits that have been performed, except with the explanation of the discrepancy between in vitro and clinical database data. Further FNBP1 downregulation in vitro appears to be anti-tumour, whereas cancer patients who have comparatively higher levels of FNBP1 in clinical databases appear to have a survival advantage.
I don't think that the text that has been added really addresses the complexity of this matter and the risk/benefit tradeoff of targeting FNBP1.
I am insisting on this because I think that having such a discrepancy unaddressed undermines the significance of the findings in the manuscript and its potential impact.
there are still some typos and incorrect verb tenses
